# Development and Validation of the Osteoporosis Knowledge, Attitude, and Behaviors Questionnaire for Female Osteoporosis Patients in Taiwan: A Mediation Model

**DOI:** 10.3390/healthcare11071023

**Published:** 2023-04-03

**Authors:** I-Chin Huang, Hui-Chuan Wu, Chih-Lung Lin, Hsiu-Hung Wang

**Affiliations:** 1College of Nursing, Kaohsiung Medical University, Kaohsiung 80708, Taiwan; 2College of Medicine, Kaohsiung Medical University, Kaohsiung 80708, Taiwan

**Keywords:** questionnaire development, osteoporosis, osteoporosis education, attitude, behaviors

## Abstract

This research develops the Osteoporosis Knowledge, Attitude and Behaviors Questionnaire (OKABQ) with the intent to evaluate the levels of osteoporosis knowledge, attitude and behavior change by developing and validating the OKABQ while establishing a mediation model of the research. A quantitative, descriptive and instrumental study was conducted in two phases: Phase I—development of the scale through Delphi Method by osteoporosis experts; and Phase II—evaluation of the validity and reliability of the scale and construction of a mediation model using SmartPLS. In Phase I, the content validity index (CVI) of the questionnaire was higher than 0.96 and the inter-rater reliability (IRR) kappa was 1.00. In Phase II, exploratory factor analysis showed that two predominating factors of attitude as a mediation were addressed by the 26-item OKABQ. The indirect effect results from the estimated model indicate that attitude mediates the relationship between knowledge and behaviors (β = 0.114, *t* = 2.627, *p* < 0.001), which is positive and statistically significant. We concluded that the OKABQ is a valid measure of osteoporosis knowledge, attitudes and behaviors for women with osteoporosis. These assessment results could assist healthcare providers in mitigating insufficiency in health education and help patients better adapt to a more active bone-healthy lifestyle.

## 1. Introduction

Osteoporosis is a major health issue in geriatrics and epidemiology, and occurrence of osteoporotic fractures lead to significant physical and economic burdens [1]. Approximately 200 million people are affected by osteoporosis, with 8.9 million fractures occurring each year worldwide [2]. In Europe, an estimated 32 million people suffered from osteoporosis in 2019, accounting for about 5.6% of the total European population over the age of 50, consisting of 25.5 million women (22.1%) and 6.5 million men (6.6%) [3]. In the 2017 National Health Survey Report, the prevalence of osteoporosis among people aged 20 to 64 was 4.4% (68.2% were women), rising to 21.5% among people over 65 years of age (72.8% were women) in Taiwan [4]. However, osteoporosis is four times more common in women than in men because women have bone loss at a younger age and this increases more rapidly in elderly women than in men [5]. Osteoporotic fractures result in significant medical costs, reduced quality of life, lost work time and productivity and increased mortality [6]. The ultimate goal of preventing and treating osteoporosis is to prevent fractures, and professionals need to provide patients with nonpharmacological and pharmacological treatments [7]. Osteoporosis is affected by various risk factors. Positive effects include adopting a healthy lifestyle that includes exercise, dietary education (appropriate calcium intake, vitamin D supplementation) and quitting smoking and drinking [8]. Patients and family members need information and knowledge from health professionals to prevent bone fragility, and multifaceted group education has shown positive impacts and could help patients enhance their ability to manage osteoporosis [9]. Briefly written educational materials could promote knowledge and belief changes but not behavioral changes in osteoporotic women [10]. The World Health Organization (WHO) has stated that using the knowledge, attitude and practice (KAP) model to conduct surveys or research is helpful for understanding the knowledge, attitudes and actual behaviors of specific groups and to communicate effectively to determine knowledge gaps, needs and problems to help plan and implement interventions [11]. Knowledge may not be the only factor that induces behavior. The result of the SEM model is that the influence of knowledge on related behaviors is indirect, and attitude is one of the moderating variables between them [12]. The KAP model is often used in public health research to explore people’s health behaviors and their changes, such as COVID-19, school education, HIV, Chronic Kidney Disease (CKD), diabetes and osteoporosis [13,14,15,16,17,18,19]. After formal education of osteoporosis, professionals may use scales to assess osteoporosis knowledge or health beliefs, although such scales only measure osteoporosis cogitation, without attitudes or behaviors under treatment [20]. One study conducted an osteoporosis questionnaire with Cronbach’s α = 0.79 that evaluated the KAP of osteoporosis among university students in Malaysia [19]. Although the questionnaire could determine perceptions of osteoporosis risk factors, it appears unsuitable for clinical tracking of post-treatment changes in osteoporosis patients.

In this study, we developed the Osteoporosis Knowledge, Attitude and Behaviors Questionnaire (OKABQ) to assess osteoporosis knowledge, attitudes and behaviors in patients with osteoporosis. The questionnaire was developed based on the experiences and perceptions of the sampled patients, physicians, nurses and the related literature [20,21,22], with the aim of (a) developing the preliminary version of the OKABQ, (b) constructing the validity of OKABQ and (c) verifying a mediation model.

## 2. Materials and Methods

### 2.1. Hypothesis

OKABQ has good psychometric properties of validity and reliability. It is used in the traditional Chinese context as an instrument for measuring the levels of osteoporosis knowledge about risk factors, attitudes toward the disease and self-efficacy for bone health behavioral change perceived in a comprehensive osteoporosis education program for osteoporotic patients. The research framework (Figure 1) indicates that the KAP model [23] was used to examine the hypothesis that attitude is a mediating variable in affecting knowledge and behaviors.

### 2.2. Study Design and Participants

A cross-sectional, descriptive instrument development study was designed. Throughout May 2019, the pilot study sample using convenience sampling was obtained via personalized contact of the principal investigator and the physician of the outpatient clinic at the hospital in Taiwan.

The inclusion criteria of this study were as follows: (a) being diagnosed with osteoporosis by a physician; (b) women aged 40 years or older; (c) being conscious and able to communicate in Mandarin Chinese or Taiwanese; and (d) agreeing to participate in the study after giving informed consent. Patients were excluded if they (a) had any mental illness or pregnancy; or (b) joined other interventional studies during the study period. Two phases were used to develop the OKABQ.

#### 2.2.1. Phase I—Initial Item Pool and Pilot Study

We used the Delphi Method to identify the concepts for osteoporosis key knowledge, attitudes and behaviors with the implementation of 2 rounds [24,25]. In the 1st round, the preliminary version of the OKABQ was drafted in Mandarin Chinese and refined by 5 osteoporosis clinical researchers and physicians with expertise in bone health issues during an in-person meeting and e-mails (from August–December 2018). Osteoporosis guidelines concerning risk factors and health issues from the World Health Organization (WHO) and the International Osteoporosis Foundation (IOF) were used as references [7,26,27,28,29,30,31,32,33,34]. A literature review and the KAP model [35] provided the structure, the component of osteoporosis disease treatment and follow-up issues. In the 2nd round, the experts evaluated the importance of each item using a Likert scale ranging from 1, “not relevant”, to 4, “highly relevant”. After informed consent was given, the patients completed the OKABQ for item analysis and assessment of the feasibility, cost, time and associated impact of the research process. The process included comments about suitability and clarity that provided drafting or added words as revised opinions.

#### 2.2.2. Phase IIa—Item Analysis, Assess Validity and Reliability of OKABQ

In this phase, we continued the study on the same patients to test the OKABQ between July 2019 to April 2020, having used convenience sampling to enroll osteoporotic patients by 2 researchers at the outpatient clinic. The questionnaire was provided as hard copies and collected at the same time, but a separate group of patients was randomized to take the test for reliability and validity collected 2 weeks later.

The 2nd version of the OKABQ included 34 items in a 1st-order confirmatory model. The Root-Mean-Square error of approximation (RMSEA) was used, with a null-hypothesized RMSEA ≤ 0.05, an alternative-hypothesized RMSEA of 0.08 and an anticipated effect size of 0.3, using normal α 0.05 and 80% power. Based on the assumptions, the minimum sample size was 170 to 291, and based on the model, it was tested as a close-fit version [36,37,38].

#### 2.2.3. Phase IIb—Mediation Analysis

In this phase, we used the data to calculate the path estimation of the model and tested the hypothesis by SEM in SmartPLS [39].

### 2.3. Data Analysis

In phase I, all statistical analyses and data entered were performed using SPSS (Version 20). The following results indicate the questionnaire was adequate in recommending a minimum S-CVI of 0.80 [40]. The readability of the items was solicited, and the validity of the questionnaire was obtained from the statistical results.

In phase IIa, the overall internal consistency of the questionnaire was analyzed as stratified Cronbach’s α, shown as adequate in recommending >0.7 as acceptable and Kuder–Richardson-20 (KR-20) was adequate in 0–1 where higher scores indicated better internal consistency [41]. Both the test and the retest were calculated by Cronbach’s α. The homogeneity and the Intraclass Correlation Coefficient (ICC) for temporal stability of the test–retest were evaluated; then, 2 Confirmatory Factor Analyses (CFA) models were used to evaluate the internal structure of the OKABQ using structural equation modeling (SEM) in SmartPLS [42] and its relationships with other variables at the latent level. Composite reliability (CR) measures internal consistency reliability with a proposed threshold value for confirmative research that values > 0.80 and must not be lower than 0.60, while average variance extracted (AVE) model can measure convergent validity with a proposed threshold value of >0.50 [43].

In phase IIb, we used CFA to test the hypothesized structure and relationships among the factors to check factor loading that allowed for estimation and removal [44,45] to construct a mediation model that followed SEM and set 5000 random subsamples in bootstrapping [43]. SEM in SmartPLS is a unique method for analyzing compound path-based models and testing a theoretical framework, ranging from 0 to 1, with higher values indicating greater explanatory power [46]. The effect sizes are represented by f^2^ [47]. Q^2^ values should be large than 0, representing that the model has predictive relevance [46,47,48].

### 2.4. Ethical Aspects

The 3-year study was approved by the Institutional Review Board (KMUHIRB-E(I)-20180317). All the participants voluntarily took part in this study and were guaranteed anonymity and confidentiality of all data.

## 3. Results

### 3.1. Phase I

The development of OKABQ was structured as follows (Figure 2):

#### 3.1.1. Preparatory and Integration Period

The first version of the OKABQ was obtained from the osteoporosis clinical researchers and experts in referring to a literature review of bone health issue; it comprised three sections related to osteoporosis, knowledge, attitude and behaviors, that included 36 items.

The second round was held with a panel of experts and a pilot study. The CVI of the overall scale and the subscale were 0.99, 1.00, 1.00 and 0.96 respectively. We tested the readability and comprehension of these items by soliciting the comments of five experts and 23 patients in the pilot study (Table 1).

Two items were rephrased and merged in the pilot test. The inter-rater reliability (IRR) of two researchers indicated a kappa of 1.00, representing perfect agreement [49].

#### 3.1.2. Construct Period

The knowledge section showed Kaiser–Meyer–Olkin (KMO) was 0.61 and the Bartlett test of sphericity was significant (*p* < 0.0001), representing non-fit by using factor analysis. The attitude and behavior sections used Exploratory Factor Analysis (EFA) with varimax rotation to extract dimensions of the OKABQ. KMO was 0.72 and 0.64 and the Bartlett test of sphericity was significant (*p* < 0.0001), indicating that these two sections were suitable for factor analysis. According to the research results, the second version of OKABQ was reduced to 34 items.

### 3.2. Phase IIa

#### 3.2.1. Descriptive Analyses

In total, 262 patients with an age range from 47.2 to 96.6 and mean age of 71.7 (SD = 9.4) participated in the study (Table 1). Most had an elementary school (29.0%) level of education, were married (57.3%), were in the postmenopausal period (92.0%) and had never smoked (98.9%), but only 26.0% had a family history of osteoporosis.

Means of total scores for the knowledge, attitude and behavior sections were scored as 47.12 (SD = 6.082), 12.08 (SD = 3.034), 18.33 (SD = 2.454) and 16.71 (SD = 2.858), respectively. Based on Kelley’s derivation [50], data was separated into high and low groups consisting of the top 27% and the lower 27% extremes of the total score. Item discrimination between high and low groups was conducted using independent sample *t*-tests. Table 2 shows significant difference (*p* < 0.001) in the total score of the scale between high and low groups, with the correlation coefficient significant between 0.130–0.762 (*p* < 0.01).

After deleting partial items, the final version of the OKABQ (Table 3) with 26 items was as follows: knowledge (16 items), ranging from 0 to 16; attitude (five items), ranging from 0 to 20 and behaviors (five items), ranging from 0 to 20. The total score ranged from 0 to 56, with higher scores indicating higher positive bone health promotion behaviors.

#### 3.2.2. Construct Validity

Table 4 shows that Average Variance Extracted (AVE) 0.52~0.84 mean convergent validity was excellent [43]. The Fornell–Larcker criterion of attitude = 0.721 is greater than those of behaviors and knowledge revealed its discriminant validity.

#### 3.2.3. Reliability

The reliability measures with Cronbach’s α of the overall scale and the sections of attitude and behaviors were 0.70, 0.71 and 0.61, respectively. The knowledge section presented internal consistency as the KR-20 coefficient was 0.78. This showed that the content validity and reliability of the overall scale were satisfactory but requires further validation with a larger study sample. In total, 28 patients completed the 30-day test–retest OKABQ for the second time, 2 of whom were excluded for loss of contact. The mean time between test and retest was 16.3 ± 3.9 (range, 10–30) days. The intraclass correlation coefficient (ICC) showed good reliability.

The construct reliability and validity for the reflective index as attitude used the SmartPLS (Table 4). Table 4 illustrates that the Cronbach’s α ranged from 0.68 to 0.82, representing a satisfactory composite reliability (CR) = 0.83~0.92, indicating good internal consistency reliability [43].

### 3.3. Phase IIb

#### Estimated Model and Mediation Analysis

The mediation analysis used SmartPLS [39]. The meaningful factors of the attitude section were paying attention to bone density state (A01) and attitudes to promote bone health (A02). The behavior section factor had five items (Figure 3).

Figure 3 reveals the path model results, indicating A01 and A02 as the reflective index. The loadings and outer weights of all items show that most were significant. Although OB-05 was not significant, it was not removed because the indicator weight = −0.257 was greater than 0.02 (Table 5) [51]. The hypothesis formulated for the mediation model was confirmed (Table 6). The f^2^ = 0.195 of knowledge to behaviors is represented as a medium effect size. The Q^2^ values = 0.034 and 0.078 indicate that the model has predictive relevance. The weak R^2^ = 0.179 depicted that attitudes caused 17.9% of the variance in behaviors [48]. The result is a mixed model (formative and reflective) of this study.

Table 7 presents the total, indirect and direct effects for the influence of knowledge on behaviors. A direct effect is the pathway estimates of a construct to behaviors. The total effect was the sum of direct and indirect effects. The results illustrate the direct effect from knowledge to behaviors (β = 0.220, *t* = 2.561, *p* = 0.010) and the indirect effect of attitudes mediating the relationship between knowledge and behaviors (β = 0.114, *t* = 2.627, *p* < 0.001) as being positive and statistically significant.

## 4. Discussion

This study developed a new scale for measuring the knowledge, attitudes and behaviors of osteoporotic patients, the OKABQ, which was modified from the KAP model, international guidelines and the literature review. We conducted a pilot study and scale validation using physicians, experts and osteoporosis patients. The current analyses have shown moderate total scores on the OKABQ, substantial construct validity and good test–retest substantiation.

The total scores showed a high level where more than 35.5% of the patients reported enhanced and positive behaviors for bone health, which might be influenced by the Hawthorne effect [52] and the undergoing of osteoporosis therapy. Clinicians should evaluate patients’ personal characteristics, preferences and unconscious judgements that affect their bone health issues. One clear indication is sample selection bias (volunteer bias) where patients willing to participate in research might possess more positive behaviors [53,54]; thus, volunteers were found to be healthier than non-volunteers in some reports.

The KAP model is one of the most popular and widely used models in medical practice, determining knowledge is the base of attitude toward behaviors [23]. The evidence indicates that education programs are effective in changing knowledge, beliefs and practice toward osteoporosis [55]. The knowledge section scored moderate levels similar to studies reporting low to moderate knowledge of osteoporosis [56,57,58]. Preventing osteoporosis is positively correlated with a higher education level in patients as having more opportunities to develop adequate knowledge and good attitudes toward preventing osteoporosis [59], illustrating that educators should consider educational level regarding enhancement of participants’ understanding.

During phase II, the OKABQ had good structure and validity. The model contains formative indicators (attitudes) and reflective indicators (behaviors). SEM is popular for indirect examination through a mediator process [60] and PLS offers a measurement model that incorporates formative and reflective indicators [61]. Among these, the R^2^ value and the path coefficients value are the main indicators for judging the quality of the model [48], where the R^2^ value was 0.163 to 0.179 and most of the path coefficients were significant. The attitude section had two principal components: paying attention to bone density state (A01) and attitudes to promote bone health (A02). The study showed declining bone mineral density (BMD) testing reasons were high cost, misconceptions about lifestyle management sufficient to prevent osteoporosis and poor awareness of the disease [56]. Although the WHO, IOF and bone health research suggest that having a fracture risk-assessment tool such as FRAX^®^ to detect bone loss is advantageous [62], BMD has a reassessed recommendation that 12–24 months after starting therapy [32], it is important to understand the degree of deterioration. Promotion of bone health is about lifestyle modification for nutrition, Vitamin D, exercise and avoiding tobacco and alcohol [63]. The most trustworthy and reliable source of health information is the clinician, but patients often feel too rushed to obtain sufficient detailed information during visits [64]; accordingly, regular health educators are required to provide more comprehensive osteoporosis education in clinical settings.

The intentions toward behavior are influenced by attitudes [65] and attitude can be the boundary between knowledge and individuals’ behaviors [66]; therefore, we planned to check prediction of behaviors from attitude, but the R^2^ values weakly depicted attitude as causing 17.9% variance in behaviors. The attitude–behavior intra-research shows that attitudes and behaviors have mediator call behavioral intentions [67]. Our finding are consistent with previous research [12].

In this study, adding behavioral intentions and merging TPB theory [35] at the beginning of the design could increase the predictive power of attitudes on behavior. Self-efficacy has been used to establish and evaluate applications designed to improve general health [68]. The key content of osteoporosis education will form the basis of behaviors, so this section presents formative indicators. These five items can provide significant measures as formative indicators of behaviors (Figure 3) as being: “having a high-calcium diet, reduction of caffeine absorption, avoiding sunbathing, avoiding overly vigorous physical activity and avoidance of falling”. Formative indicators are observed variables as causes affect a latent variable [69], meaning if any measure increases/decreases it will positively/negatively affect latent variables (but others indicators do not change); additionally, control samples from these patients may provide validity in the future phase III study.

## 5. Conclusions

The OKABQ demonstrates good construct validity, reliability, test–retest agreement and perfect inter-rater agreement. Professionals could apply the OKABQ to obtain knowledge, attitude and behavior measures of patients with osteoporosis, thereby providing more holistic educational information. Nevertheless, with the influence of knowledge on behaviors, attitude plays a mediating effect, and professionals are reminded to understand patients attitudes towards osteoporosis more deeply to improve the effect of health education and case management. Patients are also required to better understand their deficiencies in these three issues concerning osteoporosis in adjusting to a more positive bone health lifestyle.

## 6. Limitation

There are three main limitations of the study. First, longitudinal research would be more beneficial to observe changes in the knowledge, attitude and behaviors of women with osteoporosis. Second, further studies could involve male osteoporotic patients to evaluate the applicability of the OKABQ to both genders. Finally, the mediation model should be further tested using larger sample sizes.

## Figures and Tables

**Figure 1 healthcare-11-01023-f001:**
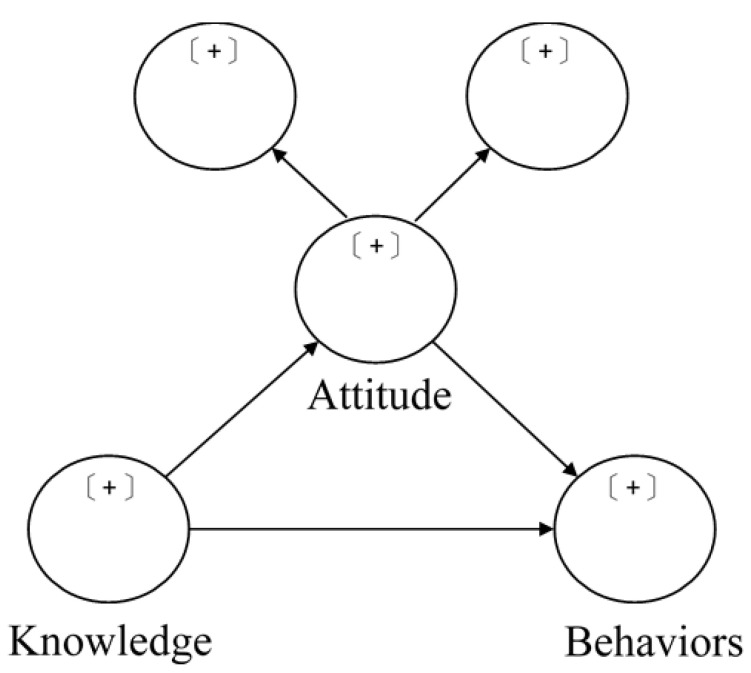
Research framework.

**Figure 2 healthcare-11-01023-f002:**
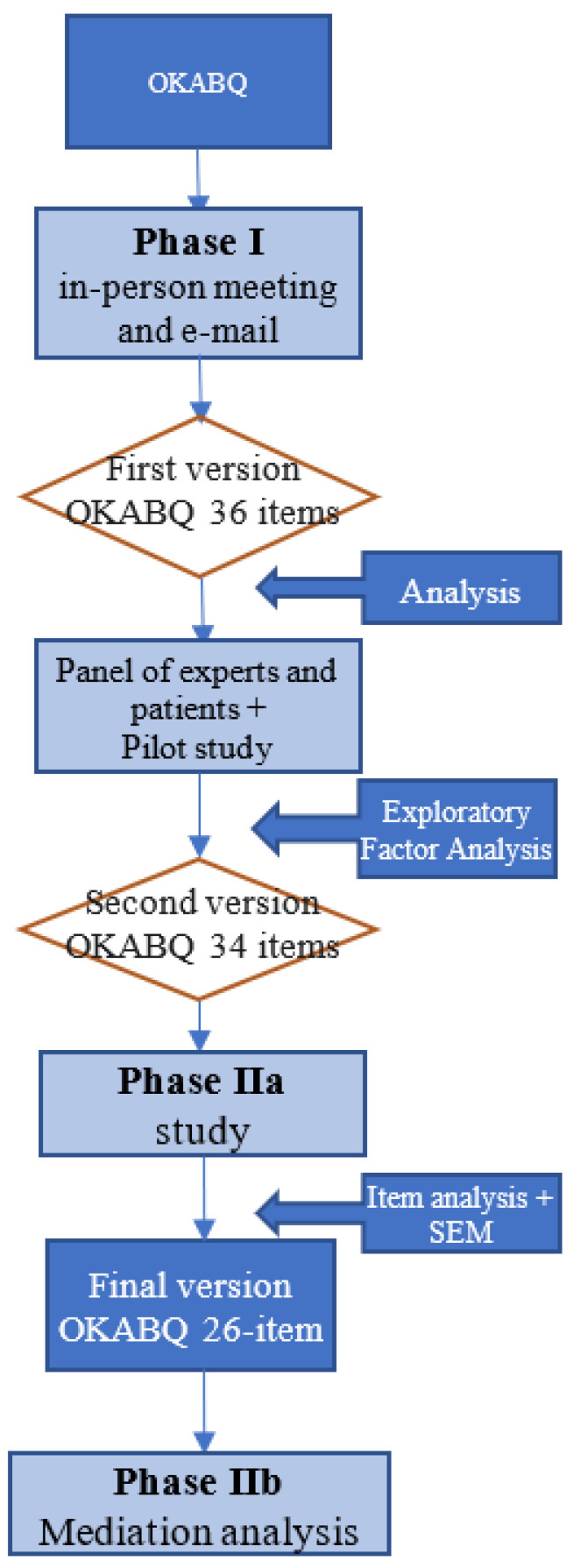
The procedure for the development of the OKABQ.

**Figure 3 healthcare-11-01023-f003:**
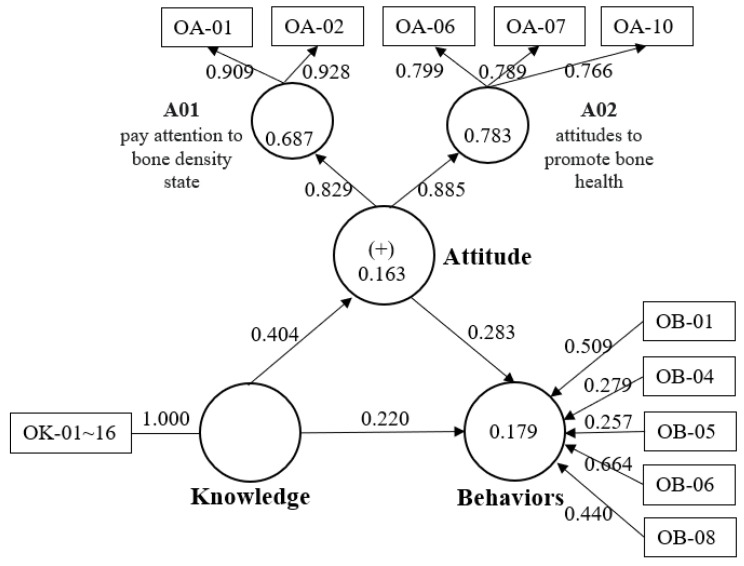
Path model of the OKHBQ using SmartPLS.

**Table 1 healthcare-11-01023-t001:** Characteristics of participants of phase I and phase II.

Characteristics	Phase I (*n* = 23)	Phase II (*n* = 262)
Age	74.2 ± 10.5	71.7 ± 9.4
Education		
Illiteracy	2 (8.7%)	31 (11.8%)
Elementary school	13 (56.6%)	76 (29.0%)
Junior high school	3 (13.0%)	43 (16.4%)
Senior high school	3 (13.0%)	67 (25.6%)
University	2 (8.7%)	35 (13.4%)
Graduate school	0 (0.0%)	10 (3.8%)
Dietary		
Mainly meat-based	13 (56.5%)	24.1 (91.9%)
Vegan	6 (26.1%)	13 (5.0%)
Vegetarian	4 (17.4%)	8 (3.1%)
Marriage		
Partner	4 (17.4%)	11 (4.2%)
Married	7 (30.4%)	150 (57.3%)
Widowed	11 (47.9%)	93 (35.5%)
Divorced	1 (4.3%)	8 (3.0%)
Menstrual condition		
Climacteric period	0 (0.0%)	4 (1.5%)
Postmenopausal period	20 (87.0%)	241 (92.0%)
Hysterectomy or ovariectomy	3 (13.0%)	17 (6.5%)
Smoking		
Never	21 (91.4%)	259 (98.9%)
Ever smoked	1 (4.3%)	2 (0.8%)
Current smoking	1 (4.3%)	1 (0.4%)
Family history of osteoporosis	1 (4.3%)	68 (26.0%)

**Table 2 healthcare-11-01023-t002:** Descriptive statistics of total scores and subscales, mean differences between high and low groups of total score (*n* = 262).

Items	Group	*n*	Mean	SD	t-Value	Correlation to Total Scale
Total scores	Low Group	70	38.81	4.810	−23.853 **	1
	High Group	71	53.03	1.320		
Knowledge scores	Low Group	70	9.03	3.097	−13.274 **	0.762 *
	High Group	71	14.39	1.368		
OA-01	Low Group	70	3.31	0.925	−5.522 **	0.460 *
	High Group	71	3.94	0.232		
OA-02	Low Group	70	3.40	0.788	−5.622 **	0.517 *
	High Group	71	3.96	0.264		
OA-06	Low Group	70	2.96	0.999	−8.733 **	0.601 *
	High Group	71	4.00	0.000		
OA-07	Low Group	70	2.73	1.089	−9.603 **	0.564 *
	High Group	71	3.99	0.119		
OA-10	Low Group	70	3.34	0.866	−5.967 **	0.499 *
	High Group	71	3.97	0.167		
OB-01	Low Group	70	2.34	1.034	−7.488 **	0.482 *
	High Group	71	3.46	0.714		
OB-04	Low Group	70	3.70	0.922	−1.689 **	0.130 *
	High Group	71	3.90	0.384		
OB-05	Low Group	70	2.47	1.259	−6.495 **	0.378 *
	High Group	71	3.62	0.781		
OB-06	Low Group	70	1.89	1.518	−9.974 **	0.593 *
	High Group	71	3.82	0.568		
OB-08	Low Group	70	3.64	0.762	−3.530 **	0.307 *
	High Group	71	3.97	0.167		

* *p* < 0.01; ** *p* < 0.001.

**Table 3 healthcare-11-01023-t003:** The final version of OKABQ.

1. Knowledge					
Items			Yes	No	Not sure
1. Osteoporosis causes bone loss and fractures easily.					
2. Only women develop osteoporosis. *					
3. Bone density does not change with age. *					
4. To sit up comfortably, choosing a lower chair or cushion is fine. *					
5. Obvious symptoms of osteoporosis will appear in the early stage. *					
6. Drinking more than three cups of black coffee, strong tea and carbonated beverages a day will increase bone loss.					
7. Vitamin D promotes calcium absorption.					
8. Healthy bone isn’t affected by sun exposure.					
9. Bone loss is accelerated in menopausal women due to a decrease in female hormones.					
10. Dietary habits are not associated with osteoporosis. *					
11. Adequate intake of calcium-containing foods can prevent osteoporosis.					
12. Speed walking or jogging can help bone formation.					
13. Once having bone loss, diet, exercise, calcium lactate or medication cannot prevent osteoporosis from getting worse. *					
14. Excessive smoking or drinking can increase bone loss.					
15. Excessively carried weight and a long-term slouched position will increase the burden on the lumbar spine.					
16. Osteoporosis is not associated with familial inheritance. *					
2. Attitude					
Items	Strongly agree	Agree	Slight agree	Disagree	Strongly disagree
1. I think it is important to have bone mass measurement regularly.					
2. I think that we should pay more attention to bone density with age.					
3. I think that increasing the time and frequency of sun exposure is helpful for bone health.					
4. I think smoking or drinking can damage bone health.					
5. I think following the health education of doctors or educator is very helpful for bone health.					
3. Behaviors					
Items	Achievement				
	100%	75%	50%	25%	0%
1. I follow a high-calcium diet (such as milk, meats and protein, dried clove fish, green vegetables) every day.					
2. I don’t drink more than three cups of black coffee, strong tea or carbonated drinks every day.					
3. I enjoy the sunshine at least 10 min every day.					
4. I have at least 30 min of physical activity three days a week.					
5. I pay attention to safety in daily life to avoid falling.					

* Reverse question.

**Table 4 healthcare-11-01023-t004:** The construct validity and reliability of the OKABQ.

	Cronbach’s α	Composite Reliability	Average Variance Extracted (AVE)
A01	0.815	0.915	0.844
A02	0.688	0.828	0.616
Attitude	0.767	0.844	0.520
Knowledge *	1.000	1.000	1.000

* As the behaviors section used informative index and the knowledge section was calculated to one continuous variable, they could not meet the calculation method.

**Table 5 healthcare-11-01023-t005:** The loadings and outer weights of all items.

Constructs	Items	Loadings	Weights	Mean	Standard Deviation (stdev)	t-Value	*p* Values
A01	OA-01	0.909		0.907	0.029	31.537 ***	0.000
	OA-02	0.928		0.929	0.013	69.672 ***	0.000
A02	OA-06	0.799		0.799	0.032	24.858 ***	0.000
	OA-07	0.789		0.788	0.039	20.021 ***	0.000
	OA-10	0.766		0.766	0.054	14.077 ***	0.000
Attitude	OA-01	0.717		0.718	0.056	12.765 ***	0.000
	OA-02	0.801		0.801	0.037	21.695 ***	0.000
	OA-06	0.715		0.713	0.046	15.474 ***	0.000
	OA-07	0.655		0.656	0.057	11.532 ***	0.000
	OA-10	0.709		0.710	0.065	10.899 ***	0.000
Behaviors	OB-01		0.509	0.483	0.148	3.44 **	0.001
	OB-04		0.279	0.258	0.134	2.078 *	0.038
	OB-05		0.257	0.249	0.191	1.343	0.179
	OB-06		0.664	0.634	0.172	3.852 ***	0.000
	OB-08		0.440	0.408	0.170	2.585 *	0.010

* *p* < 0.05; ** *p* < 0.01; *** *p* < 0.001.

**Table 6 healthcare-11-01023-t006:** Path coefficients of the model.

DV	IV	Beta	Mean	Standard Deviation (STDEV)	t-Value	*p* Values	f^2^	Q^2^	R^2^
Behaviors	Attitude	0.283	0.289	0.091	3.108	0.002	0.082	0.034	0.179
	Knowledge	0.220	0.228	0.086	2.561	0.010	0.195	-	-
Attitude	Knowledge	0.404	0.404	0.062	6.522	0.000	0.049	0.078	0.163

**Table 7 healthcare-11-01023-t007:** Total effect, indirect effect and direct effect for the model.

	Beta	Mean	(STDEV)	t-Value	*p* Values	2.5%	97.5%
Total effect							
Knowledge → Behaviors	0.334	0.346	0.069	4.812	0.000	0.202	0.461
Indirect effect							
Knowledge →Attitude → Behaviors	0.114	0.118	0.044	2.627	0.009	0.033	0.205
Direct effect							
Knowledge → Behaviors	0.220	0.228	0.086	2.561	0.010	0.054	0.383

Note: Bootstrap 5000 times, Bias-corrected percentile confidence interval.

## Data Availability

The data that support the findings of this study are available from the corresponding author, upon reasonable request.

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
