# Peer review of "Development and Validation of the Osteoporosis Knowledge, Attitude, and Behaviors Questionnaire for Female Osteoporosis Patients in Taiwan: A Mediation Model"

_healthcare, 2023, doi:10.3390/healthcare11071023_

Round 1
Reviewer 1 Report
The topic of the paper is very current, osteoporosis affects a considerable number of women around the world. A percentage of the female population affected by osteoporosis should be added depending on age and areas.
There is a need for a clearer specification of the novelties brought by this questionnaire compared to others used for a similar purpose (KAP model for example).
Author Response
Response to Reviewer 1 Comments
- Does the introduction provide sufficient background and include all relevant references? Can be improved. A percentage of the female population affected by osteoporosis should be added depending on age and areas.
Response 1: Thank you for the comments. We have redrafted the introduction on page1-2 line 27-70 to establish a clearer focus.
- There is a need for a clearer specification of the novelties brought by this questionnaire compared to others used for a similar purpose (KAP model for example)
Response 2: Thank you for the comments. We have added research materials related to KAP model in the “Introduction” section on page 2 line 48-64.

Reviewer 2 Report
This study aims to develop a questionnaire to evaluate osteoporosis knowledge, attitude and behavior change. While there are tools available to evaluate osteoporosis knowledge among patients, there is a paucity in tools that incorporate attitudes and behavior also. This study can help add to the subject area in that regard. The methodology is sound and the conclusions drawn are consistent with the evidence presented.
Author Response
Response to Reviewer 2 Comments
- While there are tools available to evaluate osteoporosis knowledge among patients, there is a paucity in tools that incorporate attitudes and behavior also.
Response 1: Thank you for the comments. We have added research materials related to KAP model in the “Introduction” section on page 2 line 48-64.
- English language and style are fine/minor spell check required
Response 2: Thank you for the comments. We have English edited the manuscript to improve the quality and provided the editing certificate.

Reviewer 3 Report
Based on a long-term study of patients with various forms of osteoporosis, the authors developed a questionnaire that is correlated with people's knowledge of osteoporosis and their lifestyles. The authors underwent the development of the form in two phases with a sufficiently large sample of observed persons, with the described variability of age, education and degree of osteoporosis. The obtained data were analytically processed and clearly point to the connection between the degree of osteoporosis and the life habits of the patients, which are associated with the knowledge of the problem.
However, the article itself and the output from the data could be better processed. If possible, improve the graphic processing of Figures for better orientation, consider the representation of data in the form of a graph and add a Methods section where the final OKABQ form would be moved together with a brief list of specific methods (separately for osteoporosis measurement and data analysis). In my opinion, it would also be beneficial to discuss the output as well as a possible informative form for young women and not only for health care providers. A control sample of knowledgeable patients would also provide valid information in the future Phase III study.
Author Response
Response to Reviewer 3 Comments
- Does the introduction provide sufficient background and include all relevant references? Must be improved.
Response 1: Thank you for the comments. We have redrafted the introduction on page1-2 line 27-70 to establish a clearer focus.
- Are all the cited references relevant to the research? Can be improved.
Response 2: Thank you for the comments. We have rechecked and revised the “References” section on page 15-18 line 447-579.
- Are the methods adequately described? Must be improved.
Response 3: Thank you for the comments. We have rechecked and revised the “Methods” section on page 2-4 line 71-153.
- Are the results clearly presented? Must be improved.
However, the article itself and the output from the data could be better processed. If possible, improve the graphic processing of Figures for better orientation, consider the representation of data in the form of a graph and add a Methods section where the final OKABQ form would be moved together with a brief list of specific methods (separately for osteoporosis measurement and data analysis).
Response 4: Thank you for the comments. We have revised the “Results” section on page 4-10 line 158-243. Also, the Figure 2 be revised to interpret the results of different methods on page 5. We were concerned that moving the final OKABQ form would make it difficult to read, so no changes have been made.
- Are the conclusions supported by the results? Can be improved.
In my opinion, it would also be beneficial to discuss the output as well as a possible informative form for young women and not only for health care providers.
Response 5: Thank you for the comments. We have revised the “Results” section on page 4-10 line 158-243.
- A control sample of knowledgeable patients would also provide valid information in the future Phase III study.
Response 6: Thank you for the comments. We have revised it on page 12 line 316-317.
- Moderate English changes required
Response 7: Thank you for the comments. We have English edited the manuscript to improve the quality and provided the editing certificate.

Round 2
Reviewer 3 Report
In my opinion, the corrections made to the article significantly increased the quality of this article.